

# The value of citizen science for ecological monitoring of mammals

Arielle Waldstein Parsons[1,2], Christine Goforth[1], Robert Costello[3] and Roland Kays[1,2]

[1] North Carolina Museum of Natural Sciences, Raleigh, NC, USA
[2] Department of Forestry & Environmental Resources, North Carolina State University, Raleigh, NC, USA
[3] Smithsonian's National Museum of Natural History, Washington, DC, USA

## ABSTRACT

Citizen science approaches are of great interest for their potential to efficiently and sustainably monitor wildlife populations on both public and private lands. Here we present two studies that worked with volunteers to set camera traps for ecological surveys. The photographs recorded by these citizen scientists were archived and verified using the eMammal software platform, providing a professional grade, vouchered database of biodiversity records. Motivated by managers' concern with perceived high bear activity, our first example enlisted the help of homeowners in a short-term study to compare black bear activity inside a National Historic Site with surrounding private land. We found similar levels of bear activity inside and outside the NHS, and regional comparisons suggest the bear population is typical. Participants benefited from knowing their local bear population was normal and managers refocused bear management given this new information. Our second example is a continuous survey of wildlife using the grounds of a nature education center that actively manages habitat to maintain a grassland prairie. Center staff incorporated the camera traps into educational programs, involving visitors with camera setup and picture review. Over two years and 5,968 camera-nights this survey has collected 41,393 detections of 14 wildlife species. Detection rates and occupancy were higher in open habitats compared to forest, suggesting that the maintenance of prairie habitat is beneficial to some species. Over 500 volunteers of all ages participated in this project over two years. Some of the greatest benefits have been to high school students, exemplified by a student with autism who increased his communication and comfort level with others through field work with the cameras. These examples show how, with the right tools, training and survey design protocols, citizen science can be used to answer a variety of applied management questions while connecting participants with their secretive mammal neighbors.

## INTRODUCTION

The monitoring and management of wildlife populations has become especially important in this age of high anthropogenic disturbance (*Kareiva, Lalasz & Marvier, 2011*). The fast pace of environmental change puts wildlife populations under increasing pressure

Corresponding author
Arielle Waldstein Parsons,
arielle.parsons@naturalsciences.org

(*Sutherland, Roy & Amano, 2015*). Ecological monitoring is a useful tool to understand and mitigate conservation concerns because it can detect changes in wildlife communities, help direct management actions and can raise the profile of conservation efforts (*Lindenmayer & Likens, 2010*; *Nichols & Williams, 2006*). For example, long term monitoring of birds in the United Kingdom and bats in the United States has led to awareness of population declines, changes in the listing of conservation status, and the development of new conservation measures (*Greenwood, 2003*; *Ingersoll, Sewall & Amelon, 2013*). However, despite the obvious benefits of monitoring programs, they remain uncommon due to the costs and logistics required, especially for wide-ranging and cryptic species.

In some cases, citizen science has provided a solution to collecting and categorizing biodiversity data on scales previously unattainable for most research teams (*Bonney et al., 2014*; *Chandler et al., 2017*; *Pereira et al., 2010*). Several types of citizen-science projects have been described based on the depth of volunteer involvement, including co-created projects, collaborative projects, and contributory projects (*Bonney et al., 2009*). The citizen science projects described in this study fit into the contributory model, in which protocols and research questions are designed by the scientists and followed by the volunteers who collect the data over wide geographic areas and long periods of time. In addition to successful use for a variety of taxa and management questions over large scales (*Barlow et al., 2015*; *Kays et al., 2016*; *Sullivan et al., 2014*), environmental monitoring by contributory citizen science adds opportunities for education and outreach, which can lead to better land stewardship by participants (*Danielsen et al., 2007*). One important challenge that must be met by every project is ensuring that citizen-collected data is of sufficient quality to be used to address scientific and management questions (*Bonter & Cooper, 2012*; *Kosmala et al., 2016*).

For mammal monitoring, this problem can be mitigated by the use of camera traps. These remotely triggered digital cameras capture a picture when an animal passes by, resulting in verifiable evidence of the animal's presence (*McShea et al., 2015*). Their relatively simple functionality, combined with the fun of looking through new animal pictures, make them ideal for use by non-scientists. Projects working with citizen scientist-run camera traps are able to collect large amounts of geo-referenced, verifiable data; however, the logistics of expert verification and management of this large number of photographs presents a new problem. eMammal is a software system developed to address this challenge by providing a workflow to facilitate camera trap research conducted by citizen scientists (*McShea et al., 2015*). The eMammal system includes software for viewing, tagging, and uploading photographs, an expert review tool to ensure data quality, a repository to store approved data, and a website for project and volunteer management, data access and analysis (*McShea et al., 2015*). eMammal also provides a set of recommended protocols and a minimum metadata standard to make data comparable across studies. All approved data are stored in the Smithsonian's repository for scientific data, which provides secure long-term storage and an avenue for making data publicly accessible.

Here we present two studies to show the value of citizen scientists in gathering data and influencing management actions while gaining positive personal results for participants. We used citizen scientist-run camera traps and the eMammal software system to monitor mammal activity (i.e., intensity of use derived from detection rate (the number of detections

of a given species divided by the total number of camera-nights, hereafter "DR") and occupancy) and answer specific management questions. We used a contributory model of citizen-science for both studies wherein scientists and managers formulated the research questions and survey design and volunteers collected the data. The management goal of our first study was to quantify black bear (*Ursus americanus*) activity inside Carl Sandburg Home National Historic Site (hereafter Sandburg Home), Flat Rock, North Carolina, United States over two months. The frequency of human-bear encounters has increased in western North Carolina and growing numbers of communities now live with black bears. Managers of Sandburg Home have been considering management actions to control a perceived overabundant bear population, however evidence for this overpopulation is anecdotal and based on visitor reports of bear sightings. Thus, prompted by concerns over visitor and bear safety, park managers called for this study to determine if they truly had a problem of overabundance. Since the park is small and embedded within a close-knit community, and bears are a particularly polarizing species (e.g., both charismatic and potentially dangerous), managers wanted community involvement to foster understanding and contribute to future management planning for bears in the area. To determine whether Sandburg Home and surrounding neighborhoods had overly high bear activity (an indicator of overabundance), we surveyed bears within Sandburg Home and simultaneously engaged local homeowners to survey surrounding private properties. We compared bear activity inside and outside of Sandburg Home and with other sites throughout the region.

Our second example is a long-term continuous survey of wildlife at an environmental education center, Prairie Ridge Ecostation (hereafter Prairie Ridge), Raleigh, North Carolina, United States. Prairie restoration began at Prairie Ridge in 2004 by removal of fescue and Johnson grass and planting of native tallgrass prairie species, followed by spring burns and mowing of sections of the prairie on a three-year rotation to promote plant diversity (*Yelton, 2007*). The effect of active prairie restoration on the mammal community at Prairie Ridge has never been evaluated, despite prairie conversion beginning more than a decade ago. Prairie restoration is expected to affect species differently, benefiting species adapted to early successional habitat that is increasingly scarce over the region (*Askins, 2001*). Evaluating the impact of prairie restoration on the mammal community (via measures of diversity, seasonal patterns, species interactions and activity (i.e., DR and occupancy) is important for future management actions related to prairie conversion at the site. The education and outreach goals of this study included involving visitors in scientific research, improving knowledge of local mammal species and improving understanding of the benefits of urban wildlife habitat. To achieve these goals, center staff incorporated the camera traps into ongoing educational programs and involved visitors with camera setup and picture review.

## MATERIALS AND METHODS

### Study sites
#### *Sandburg Home*
Sandburg Home is a 1.07 km$^2$ property located in Flat Rock, western North Carolina, United States (Fig. 1). The majority of the property is mature forest which includes 8 km

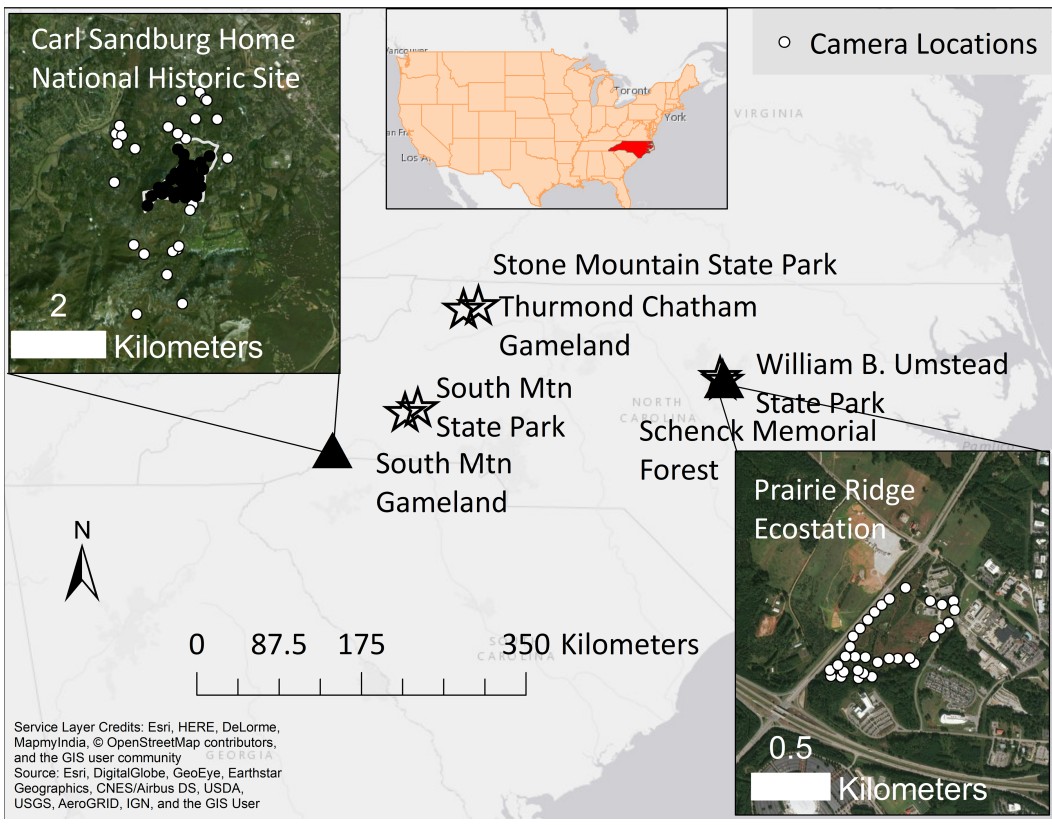

**Figure 1** **A map of our two study sites (triangles) and comparative sites (stars) in North Carolina, United States.** Black dots at the Carl Sandburg Home National Historic Site inset map show sites inside the park where camera traps were run by volunteers, while white dots show where citizens ran cameras on private property. White dots at the Prairie Ridge inset map show stations where volunteers ran camera traps over three years, rotating them between sites every four weeks. Comparative sites were sampled by citizen scientists in 2011–12 as part of an earlier study (*Kays et al., 2016*). Service layer credits: Esri, HERE, DeLorme, MapmyIndia, ©OpenStreetMap contributors and the GIS user community. Image data: Esri, DigitalGlobe, GeoEye, Earthstar Geographics, CNES/Airbus DS, USDA, USGS, AeroGRID, IGN, and the GIS User.

of hiking trails. The property was originally owned by poet Carl Sandburg and was opened as a National Historic Site in 1974. The southwestern corner of the property contains pastures, ponds, and a total of fifty structures, including the Sandburg's residence and goat barn. Sandburg Home is open to visitors year-round.

### Prairie ridge

Prairie Ridge is a 0.16 km$^2$ property located in Raleigh, central North Carolina, United States (Fig. 1). The property includes both forested and open habitat but, in contrast to Sandburg Home, is mostly open. The majority of open habitat was formerly cow pasture which has been converted to Piedmont prairie. Piedmont prairie is an open savannah-like habitat, resulting from regular fire, land clearing and grazing (*Davis et al., 2002*). While the original pre-settlement extent of Piedmont Prairie is unknown, accounts from early settlers suggest Piedmont prairie was widespread throughout central North Carolina
(*Juras, 1997*). Early successional habitats like prairies are some of the most diverse habitats in North America but they have declined in the last 200 years, leading to declines in shrub and grassland specialists (*Askins, 2001*). Indeed, remnant prairies are all that are left in North Carolina (*Barden, 1997*). Prairie Ridge is open to visitors year-round and provides numerous educational opportunities through its outdoor classroom, interpretive trails, nature playspace, sustainable building features and native tree arboretum.

## Camera trap surveys
### Sandburg home

From August-October 2015, 30 Reconyx RC55 cameras (Reconyx, Inc. Holmen, WI, USA) were set inside the Sandburg Home grounds by eMammal staff and student volunteers from Haywood Community College and North Carolina State University. Cameras were equipped with an infrared flash and attached to trees 40 cm above the ground and ran for one month without being checked to limit human scent influencing animal activity. Camera sites within Sandburg Home were chosen at random, to get as representative a sample as possible, and spaced at least 200 m apart. Twenty-seven additional Reconyx PC800 cameras were set by volunteers in the surrounding neighborhoods. Volunteers were provided with all necessary equipment, including cameras, memory cards, batteries and camera locks. Cameras recorded five photographs per trigger, at a rate of 1 frame/s, re-triggering immediately if the animal was still in view. For analysis we grouped consecutive photos into sequences if they were <60 s apart, and used these sequences as independent records for subsequent analysis (*Kays et al., 2016*). We grouped data into daily detection/non-detection for each species to use in occupancy modeling.

### Prairie ridge

From November 2013 to June 2016, eight cameras were rotated around 32 fixed stations every four weeks, completing a full rotation every four months for a total of six full rotations. Stations were spaced at least 200 m apart. Half of the camera stations were placed in the forest fragments around the edges of the property and the other half were located in open areas adjacent to areas of piedmont prairie. Volunteers and center staff used Reconyx (PC800, and PC900; Reconyx, Inc. Holmen, WI, USA) and Bushnell (Trophy Cam HD; Bushnell Outdoor Products, Overland Park, KS, USA) camera traps equipped with an infrared flash and attached to trees 40 cm above the ground. Cameras were left for four weeks before moving them to new locations, without being checked to limit human scent influencing animal activity. Bushnell camera sensitivity was initially set to high at the beginning of the study, but large amounts of empty frames in grassy areas prompted us to reduce the sensitivity to medium beginning in spring 2014. All cameras were subsequently switched from Bushnell cameras to Reconyx cameras set with high sensitivity in winter 2014. We tested for differences in overall DR (i.e., all animals combined) between the same seasons in each sensitivity period (Fall-Winter 2013, Spring-Fall 2014 and Winter 2014-Spring 2016; Fig. S1) using a Wilcoxon signed-rank test in Program JMP (SAS, Cary, NC, USA). Cameras recorded three or five photographs per trigger (Bushnell and Reconyx respectively), at a rate of 1 frame/s, re-triggering immediately if the animal was still in view. For analysis we grouped consecutive photos into sequences if they were <60 s apart, and used these
sequences as independent records for subsequent analysis (*Kays et al., 2016*). We grouped data into daily detection/non-detection for each species to use in occupancy modeling.

## Volunteer recruitment and training
### Sandburg home

We recruited student volunteers by contacting professors and student groups and recruited neighborhood volunteers by distributing flyers and through existing contacts at Sandburg Home. All field activities at Sandburg Home were approved by the US National Park Service under permit #CARL-2017-SCI-0002. We chose homes among respondents such that clustering of sample sites was minimized and proximity to Sandburg Home was within 1.5 km. Most of our neighborhood participants were adults, although some minors did participate with the supervision of a parent. All volunteers who helped set cameras for the project were trained either in person or online to ensure that all camera protocols were standardized. Trainings were comprehensive and included how to use a GPS enabled device, how to setup and use a camera trap, how to use the eMammal software and how to identify mammal species.

### Prairie ridge

Most repeat volunteers were recruited through the NC Museum of Natural Sciences' volunteer program and were the primary participants in the monthly camera movements, data processing, and data uploads. Other volunteers were recruited through public mammal program offerings at Prairie Ridge that incorporated camera movements and image review as part of the lesson. Several grade school groups participated, as well as multiple college-aged interns who helped move cameras and process data. Finally, several teen volunteers approached the program leaders directly and were incorporated into the program. These included both high achieving and special needs high school students. We surveyed primary participant volunteers after conducting camera trapping at Prairie Ridge to evaluate their experience and any impact on their attitudes towards wildlife. We administered a similar control survey via Twitter to residents in the same three-county area as Prairie Ridge. Surveys were short (<10 questions) and involved a combination of Likert-scale responses and short answers (Table S1). All survey protocols included written consent and were approved by the North Carolina State University Institutional Review Board (#11902). We assessed significant differences in responses between the control group and Prairie Ridge participants using a Fisher's exact test, which is appropriate for small sample sizes, in R (*R Development Core Team, 2008*) via R Studio (*RStudio Team, 2015*).

## Data collection and verification

Volunteers used the custom eMammal desktop software application to manually identify animal pictures and upload the data to the eMammal cloud storage location (see *McShea et al., 2015* for details). The volunteers used a field guide of mammals that could be found in their geographic area to tag species in photos. Experts in mammal identification subsequently reviewed each photo identified by the volunteers using the eMammal web-based data review tool. Where necessary, identifications made by the volunteers were corrected to ensure photo identification was correct upon entering permanent storage in

the Smithsonian digital data repository. Past studies using this system have noted success rates for volunteer identification of over 90% (*McShea et al., 2015*).

## Analyses

We used DR to compare the relative activity levels of each species (i.e., intensity of use). Since DR does not account for imperfect detection, we also estimated occupancy probabilities as a complementary metric to measure activity levels and habitat associations. These two metrics tend to be highly correlated and are both considered measures of relative abundance, not true abundance (*Parsons et al., 2017*). Specifically, we used the single season occupancy modeling framework of *MacKenzie et al. (2006)* and estimated detection probability ($p$), defined as the probability of detecting an occurring species at a camera site, and occupancy ($\psi$), defined as the expected probability that a given camera site is occupied, for each species. We ran a single model for each species, modelling $p$ using detection distance, the farthest distance away the camera would trigger on a person, measured when each camera was set to account for differences in terrain and vegetation and using categorical covariates to predict $\psi$ (i.e., open or forest, protected area). We ran our models using the RMark package (*Laake, 2015*) in R (Version 3.4.0, *R Development Core Team, 2008*) via R Studio (Version 1.0.143, *Team R, 2015*). We calculated Shannon diversity using package iNext (*Hsieh, Ma & Chao, 2016*) in R (Version 3.4.0, *R Development Core Team, 2008*) via R Studio (Version 1.0.143, *RStudio Team, 2015*). We compared DR and occupancy at Prairie Ridge and Sandburg home and diversity at Prairie Ridge to nearby sites from a previous study (*Kays et al., 2016*; *Parsons et al., 2016*), assessing differences in total DR (all detected species combined) using the Wilcoxon signed-rank test for nonparametric comparisons in Program JMP (SAS, Cary, NC, USA) and differences in occupancy and diversity using confidence interval overlap. Data from the comparison sites are freely available and were downloaded from eMammal.org ("Parsons Case Studies" by Arielle Parsons, licensed under CC Attribution-NonCommercial-ShareAlike 4.0 International: https://creativecommons.org/licenses/by-nc-sa/4.0/, DOI: 10.25571/01) and are included in the raw dataset associated with this publication. We tested seasonal correlations in DR and occupancy between species using a Pearson's correlation coefficient in JMP (SAS, Cary, NC, USA). Most cameras were not located on hiking trails with the exception of two at Prairie Ridge.

## RESULTS

### Monitoring and management

#### Sandburg home

Over 1,591 camera-nights we collected 3,252 detections (Table 1) of 15 wildlife species. Sandburg Home total mammal DR (all species combined) was not significantly different than any nearby protected areas except South Mountains Gameland ($p = 0.03$) (Fig. 2, Table 2), although these sites were sampled in a different year which could affect DR. We noted significantly lower white-tailed deer (*Odocoileus virginianus*) and significantly higher eastern gray squirrel *(Sciurus carolinensis)* DR at Sandburg Home compared to adjacent sites (i.e., all pairwise $p < 0.05$) (Fig. 2, Table 2). Sandburg Home had similar levels of bear

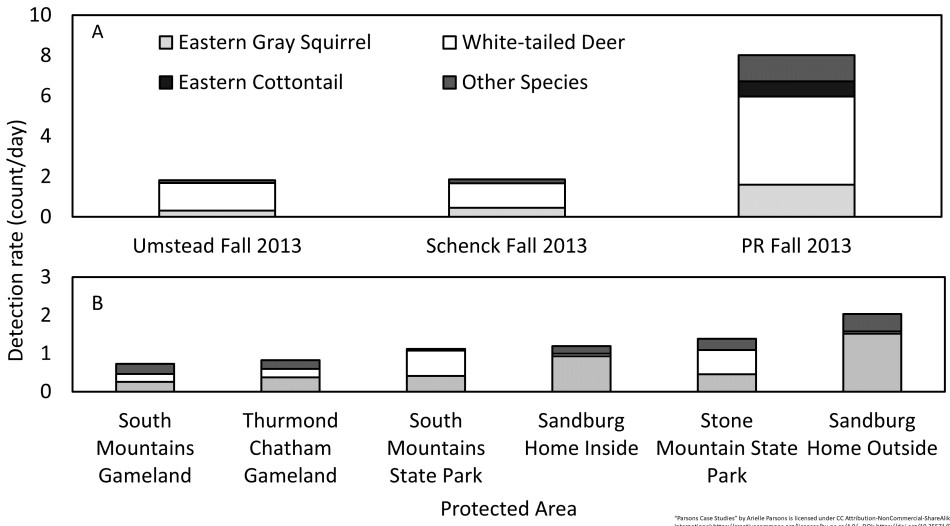

**Figure 2** **Diversity and relative abundance of wildlife species detected by camera traps.** Prairie Ridge Ecostation (A) was sampled in Fall 2013 and compared to two nearby sites sampled during the same time-frame and Carl Sandburg Home National Historic Site (B) was sampled in Fall 2015 and compared to four nearby sites sampled in Fall 2012.

**Table 1** **Research objective, sampling effort, survey area and volunteer effort for camera trap surveys at Carl Sandburg Home National Historic Site (2015) and Prairie Ridge Ecostation, North Carolina, USA (2013–2016).**

|  | Carl Sandburg Home NHS | Prairie Ridge Ecostation |
|---|---|---|
| Research objective | Compare bear activity inside and outside the park and with other regional parks | Survey diversity of park and monitor seasonal changes in animal activity. |
| Camera locations | 55 | 170 |
| Camera nights | 1,591 | 5,968 |
| Animal detections | 3,252 | 41,393 |
| Sample period | 3 months | 2 years |
| Survey area | 9,100 km$^2$ | 325 km$^2$ |
| Number of camera traps used | 50 | 8 |
| Camera trappers | Project staff ran cameras in park, volunteers ran cameras outside park | Visitors to the park ran cameras as part of educational programs |
| Total volunteer hours | 105 | 510 |

**Notes.**

activity when compared to adjacent sites but had significantly higher bear activity than nearby South Mountains State Park, which was significantly lower than all other sites (i.e., all pairwise $p < 0.05$) (Fig. 3, Table 2). Bear occupancy at Sandburg Home was similar to other sites except Stone Mountain State Park which was significantly lower (Fig. 3).

### Prairie ridge

Over 5,968 camera-nights we collected 41,393 detections (Table 1) of 14 wildlife species including 10 resident species detected regularly throughout the study and four "visiting

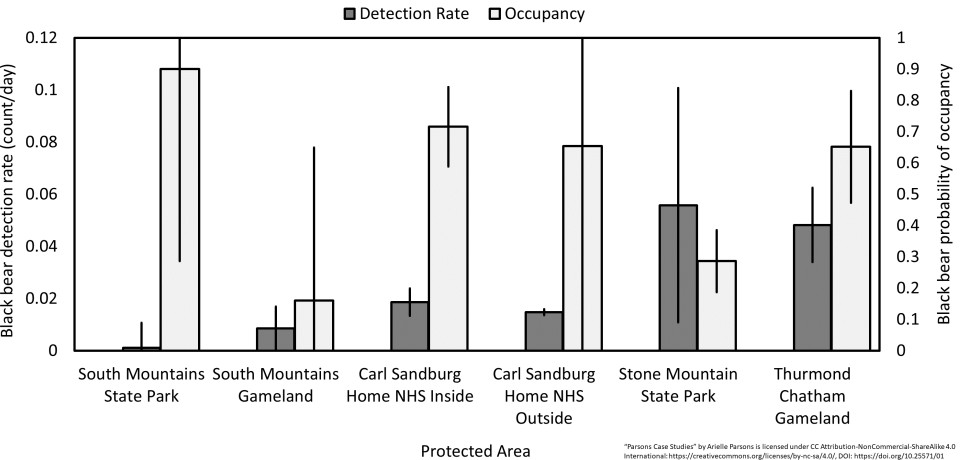

**Figure 3** **Black bear detection rate (count/day) and probability of occupancy.** Data were taken from camera traps at Carl Sandburg Home National Historic Site (inside and outside), Flat Rock, NC, USA from August to October 2015 and compared with nearby protected areas sampled for three months in 2012 and 2013 (error bars are standard error).

**Table 2** **Differences in American black bear, white-tailed deer and eastern gray squirrel detection rate (count/day) between Sandburg Home (Inside) and nearby protected areas sampled with camera traps in Fall 2015 and 2012 respectively.**

| | American black bear | | | | White-tailed deer | | | | Eastern gray squirrel | | | |
|---|---|---|---|---|---|---|---|---|---|---|---|---|
| Area | Mean | SE | Z | p | Mean | SE | Z | p | Mean | SE | Z | p |
| SHI | 0.0 | 0.0 | NA | NA | 0.1 | 0.0 | NA | NA | 0.9 | 0.2 | NA | NA |
| SHO | 0.0 | 0.0 | 0.2 | 0.9 | 0.1 | 0.1 | 0.3 | 0.1 | 1.4 | 0.5 | 0.5 | 0.6 |
| SMG | 0.0 | 0.0 | 0.3 | 0.2 | 0.2 | 0.0 | 3.0 | **0.0** | 0.3 | 0.1 | −3.4 | **0.0** |
| SMSP | 0.0 | 0.0 | −2.2 | **0.0** | 0.6 | 0.1 | 5.0 | **0.0** | 0.4 | 0.1 | −2.5 | **0.0** |
| STM | 0.1 | 0.0 | −0.2 | 0.8 | 0.6 | 0.1 | 5.4 | **0.0** | 0.4 | 0.1 | −2.3 | **0.0** |
| TCG | 0.1 | 0.0 | 1.7 | 0.1 | 0.3 | 0.1 | 3.3 | **0.0** | 0.4 | 0.1 | −3.2 | **0.0** |

**Notes.**

Z test statistics and associated p-values are the result of nonparametric Wilcoxon signed-ranks comparisons and those in bold represent a significant difference at the 0.05 level.

"Parsons Case Studies" by Arielle Parsons is licensed under CC Attribution-NonCommercial-ShareAlike 4.0 International: https://creativecommons.org/licenses/by-nc-sa/4.0/, DOI: 10.25571/01.

SHI, Sandburg Home Inside; SHO, Sandburg Home Outside; SMG, South Mountains Gameland; SMSP, South Mountains State Park; STM, Stone Mountain State Park; TCG, Thurmond Chatham Gameland.

species" that we detected rarely and episodically, suggesting they were just passing through the preserve (Fig. 4). Changing trigger sensitivity did not significantly change DR over all species, however we did note differences between camera models, though only in Spring and Summer (Table 3). Since camera model did not significantly change DR in the Fall, which is the only season shared between the three periods, we conclude that differences in detection rate are more likely a result of natural factors rather than significant differences in detection probability. Shannon diversity (Fig. S2, Table 4) and total mammal DR at Prairie Ridge were significantly higher than other more heavily forested sites nearby sampled during the same timeframe (Fall 2013) (DR: $p < 0.001$ for each pairwise comparison). The high DR was driven by white-tailed deer, eastern cottontail and eastern gray squirrel (Fig.

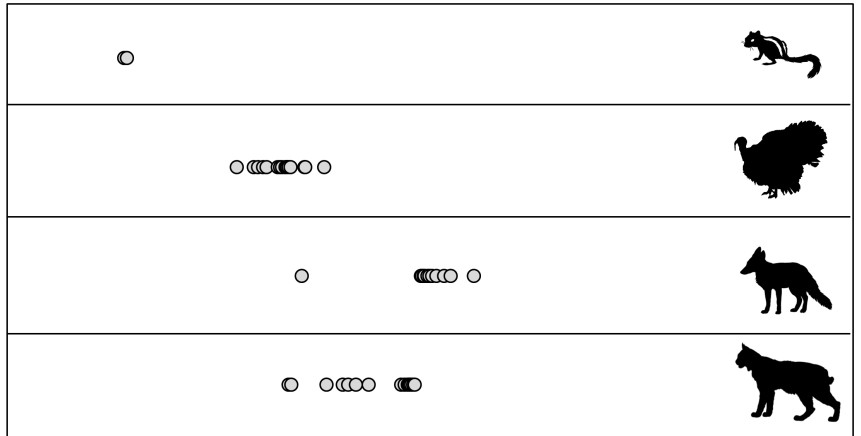

5/6/2013  11/22/2013  6/10/2014  12/27/2014  7/15/2015  1/31/2016  8/18/2016

Date

**Figure 4** **Episodic visitation of non-resident species at Prairie Ridge Ecostation.** Data were taken from camera traps run between May, 2013 and August, 2016. Species from top to bottom are eastern chipmunk, wild turkey, red fox and bobcat.

**Table 3** **A comparison of the impact of trigger sensitivity changes on overall DR (i.e., all species) from camera traps run at Prairie Ridge Ecostation, Raleigh, NC, USA from 2013–2016.** Sensitivity changes occurred at the beginning of Spring 2014 (High sensitivity to Medium sensitivity) and Winter 2014 (Bushnell brand cameras switch to Reconyx).

| Sensitivity change | Months | Mean difference | SE of Difference | Z | p |
|---|---|---|---|---|---|
| High to Medium | Fall 13–Fall 14 | −4.1 | 3.2 | −1.3 | 0.2 |
| Bushnell to Reconyx | Fall 14–Fall 15 | 0 | 2.9 | 0 | 1 |
| Bushnell to Reconyx | Spring 14–Spring 15 | −9.8 | 3.6 | −2.6 | **0.0** |
| Bushnell to Reconyx | Spring 14–Spring 16 | −9.5 | 3.7 | −2.6 | **0.0** |
| Bushnell to Reconyx | Summer 14–Summer 15 | −12.6 | 3.8 | −3.3 | **0.0** |

**Notes.**

Statistics are the result of pairwise Wilcoxon signed-rank comparisons with bolded $p$-values showing significance at the 0.05 level. "Parsons Case Studies" by Arielle Parsons is licensed under CC Attribution-NonCommercial-ShareAlike 4.0 International: https://creativecommons.org/licenses/by-nc-sa/4.0/, DOI: 10.25571/01.

2). We noted significantly higher eastern cottontail occupancy in Prairie Ridge compared to nearby sites, however total mammal occupancy was not significantly different, possibly because white-tailed deer occupied all sites, reaching the estimator asymptote and masking trends in relative abundance between sites (Fig. S3).

Trends over time showed a concurrent drop in gray fox (*Urocyon cinereoargenteus*) when coyote (*Canis latrans*) detections rose at the beginning of the study (Fig. 5). Other species like woodchuck (*Marmota monax*) showed clear seasonal patterns (Fig. 6, Fig. S4). Eastern cottontail, white-tailed deer, Virginia opossum (*Didelphis virginiana*) and coyote were detected more often and all species except Virginia opossum had higher occupancy probabilities in open habitats adjacent to prairie restoration. All other resident species were detected equally or more in forested compared to open habitats, with the same trend
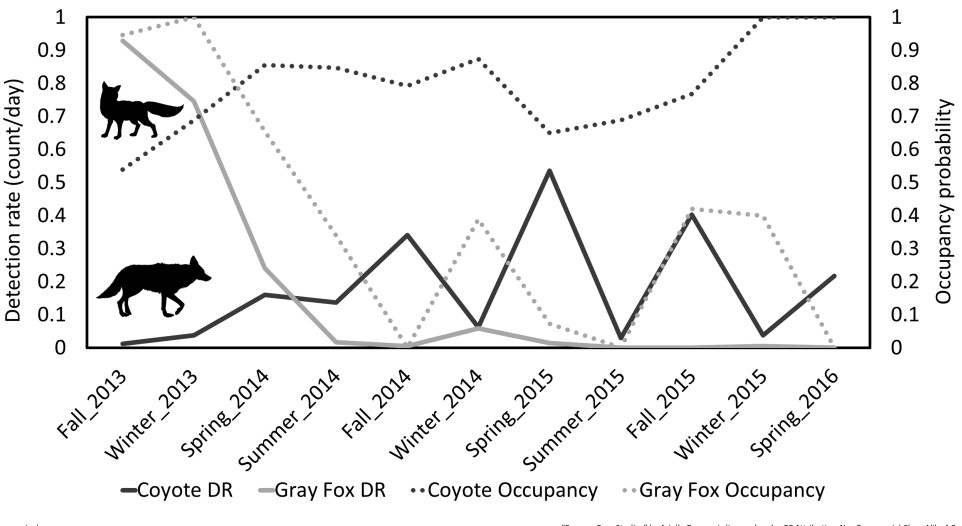

www.pixabay.com
http://www.supercoloring.com/silhouettes/, Bob Comix

"Parsons Case Studies" by Arielle Parsons is licensed under CC Attribution-NonCommercial-ShareAlike 4.0
International: https://creativecommons.org/licenses/by-nc-sa/4.0/, DOI: https://doi.org/10.25571/01

**Figure 5** **Seasonal patterns of detection rate (total count/day) and occupancy for coyote and gray fox at Prairie Ridge Ecostation, Raleigh, NC, USA.** Data were taken from cameras traps run between 2013 and 2016.

**Table 4** **Shannon diversity index estimates from rarefaction analysis between Prairie Ridge and nearby comparison sites. Data are taken from camera traps run in Fall 2013.**

| Site | Estimator | SE | 95% CI Upper | 95% CI Lower |
| --- | --- | --- | --- | --- |
| Prairie Ridge | 1.2 | 0.0 | 1.2 | 1.2 |
| Schenck | 1.2 | 0.0 | 1.2 | 1.2 |
| Umstead | 1.1 | 0.0 | 1.1 | 1.2 |

**Notes.**
"Parsons Case Studies" by Arielle Parsons is licensed under CC Attribution-NonCommercial-ShareAlike 4.0 International:
https://creativecommons.org/licenses/by-nc-sa/4.0/, DOI: 10.25571/01.

in occupancy except for woodchuck which had slightly higher occupancy in open areas (Fig. 7, Fig. S5). We noticed a similar longitudinal pattern in DR and occupancy between Virginia opossum, northern raccoon (*Procyon lotor*) and domestic cats (*Felis catus*) and found significant DR correlations between cats and both Virginia opossum and northern raccoon in the winter and fall months (Fig. 6, Figs. S4, Table 5).

## Education and outreach
### Sandburg home
In total, 42 citizen scientists participated in this project, contributing 285 volunteer hours. The ecological results of this study showed participants that their local bear population was typical of the region. The study results and photographs are being used to develop site-specific materials that assist visitors to Sandburg Home in understanding bear presence and behavior at the site, thus reducing the risk of unsafe bear encounters. Outreach via a news article in the Hendersonville, NC Times-News local media broadened the audience to neighboring communities that may be experiencing similar concerns about bear activity.

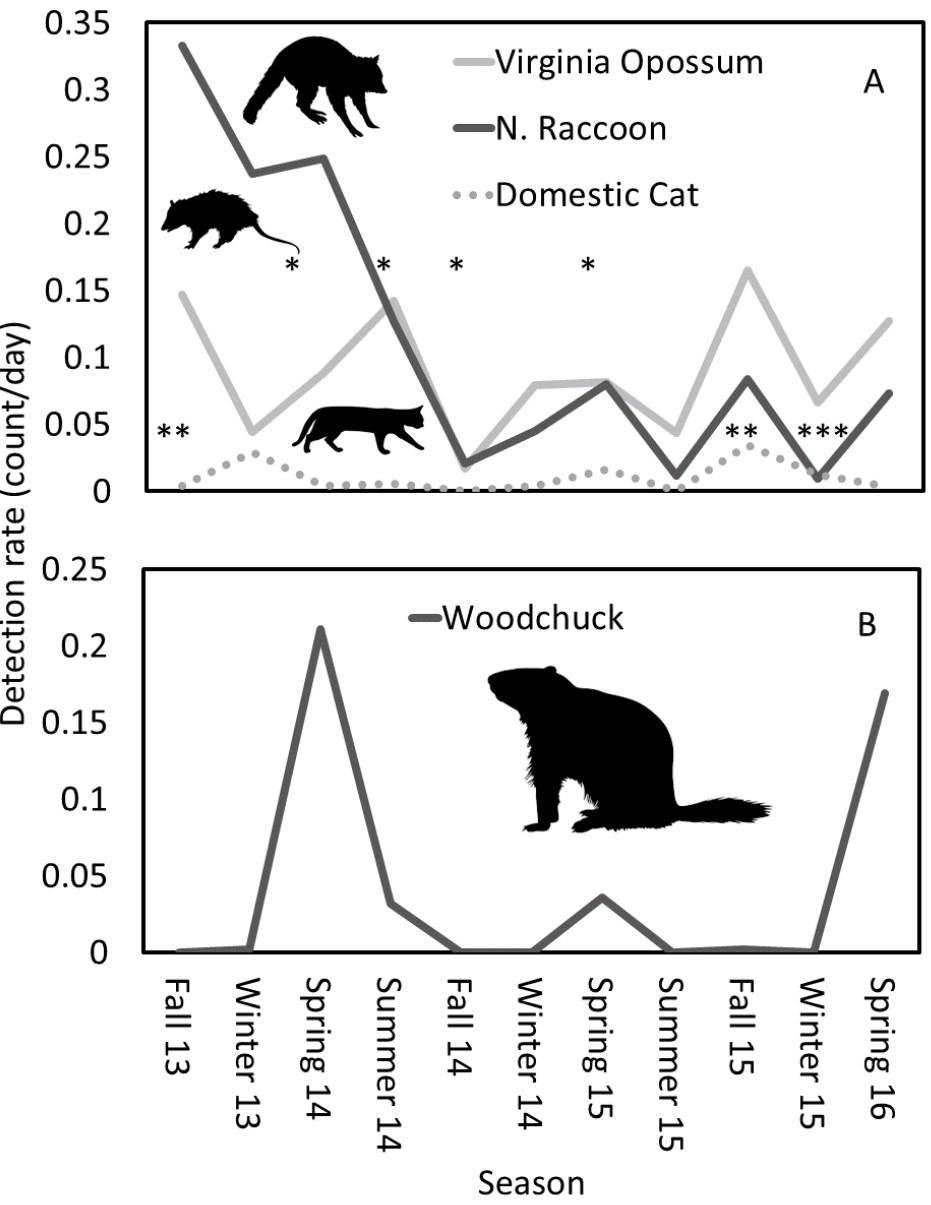

**Figure 6** **Seasonal patterns of detection rate for (A) northern raccoon, Virginia opossum and domestic cat and (B) woodchuck at Prairie Ridge Ecostation, Raleigh, NC, USA.** Data were taken from camera traps run between 2013 and 2016. * denotes significant correlations between Virginia opossum and northern raccoon, ** denotes significant correlations between domestic cats and Virginia opossum and *** denotes significant correlations between domestic cats and northern raccoons. "Parsons Case Studies" by Arielle Parsons is licensed under CC Attribution-NonCommercial-ShareAlike 4.0 International.

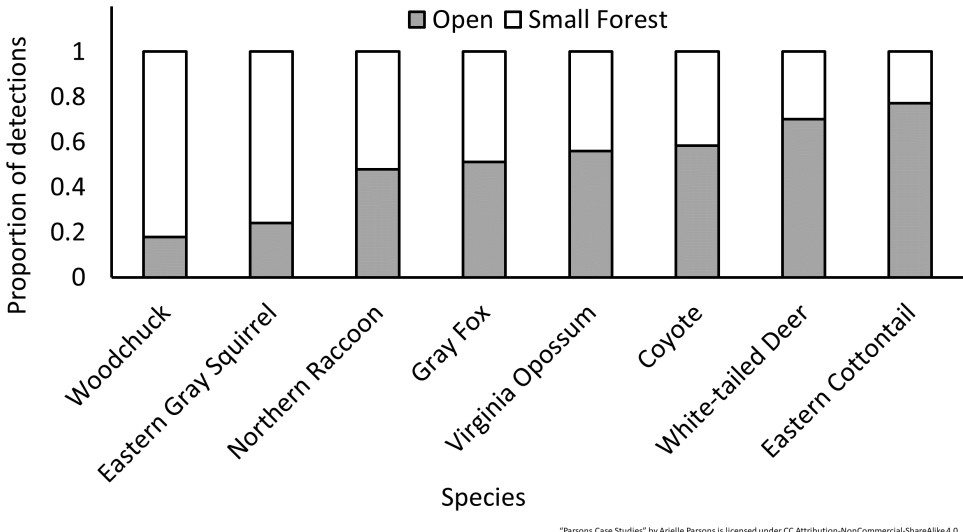

**Figure 7** **The proportion of detections in open and forest habitats for the most common resident species at Prairie Ridge Ecostation, Raleigh, NC.** Data were taken from camera traps run from 2013 to 2016.

Local participants became more aware of bear activity and reached out to their community with self-published newsletters edited for accuracy by eMammal staff, further widening the scope of project influence and information sharing.

### Prairie ridge

Overall, 531 volunteers ran cameras or uploaded pictures for this project over two years, however most of these were large groups that visited only once or twice. Thirty-five of these volunteers worked with the cameras on a regular basis (i.e., the primary participants in the monthly camera movements, data processing, and data uploads) and were invited to take our survey. Out of nine participants who completed our survey, one reported connections with researchers at Prairie Ridge as their greatest benefit, two reported benefitting from participating in real research and six reported benefitting from an enhanced awareness of wildlife. Prairie Ridge participants expressed significantly greater interest in wildlife compared to the control group (Fig. S6, Table 6). All Prairie Ridge participants reported improved understanding of the value of urban habitat fragments such as Prairie Ridge, compared to five out of six participants in the control survey who expressed caring about whether wildlife could live in urban areas, although this difference had only marginal significance (Fig. S6, Table 6). Prairie Ridge participants reported becoming more interested in observing species with camera trap footage rarely seen firsthand, particularly coyotes, deer, rodents, groundhogs, foxes and ducks. Indeed, seven out of nine participants claimed to like coyotes after camera-trapping at Prairie Ridge compared to three out of six from the control survey, though this difference was not statistically significant (Fig. S6, Table 6). Six out of the nine Prairie Ridge volunteers surveyed reported becoming more comfortable with the Prairie Ridge environment and interactions with researchers after participating

**Table 5  Pearson correlation coefficients (ρ) and associated p values for pairwise comparisons of detection rate and occupancy from camera traps between domestic cat, northern raccoon and Virginia opossum at Prairie Ridge Ecostation, Raleigh, NC over 11 seasons (2013–2016).** Significant correlations are denoted in bold.

| Season | Raccoon-Cat | | Possum-Cat | | Possum-Raccoon | |
|---|---|---|---|---|---|---|
| | ρ | p | ρ | p | ρ | p |
| Detection Rate | | | | | | |
| Fall 13 | 0.2 | 0.4 | 0.9 | **0.0** | 0.3 | – |
| Winter 13 | −0.1 | 0.7 | −0.1 | 0.6 | −0.1 | 0.6 |
| Spring 14 | −0.1 | 0.7 | −0.2 | 0.5 | 0.8 | **0.0** |
| Summer 14 | 0.3 | 0.2 | 0.1 | 0.7 | 0.7 | **0.0** |
| Fall 14 | – | – | – | – | 0.8 | **0.0** |
| Winter 14 | −0.2 | 0.6 | 0.0 | 1.0 | 0.2 | 0.4 |
| Spring 15 | 0.3 | 0.3 | 0.3 | 0.4 | 0.9 | **0.0** |
| Summer 15 | – | – | – | – | −0.1 | 0.8 |
| Fall 15 | 0.4 | 0.1 | 0.6 | **0.0** | 0.2 | 0.5 |
| Winter 15 | 0.9 | **0.0** | 0.3 | 0.5 | 0.2 | 0.6 |
| Occupancy | | | | | | |
| Fall 13 | 0.5 | 0.7 | 0.3 | 0.8 | 1.0 | 0.1 |
| Winter 13 | 1.0 | **0.0** | 1.0 | 0.2 | 1.0 | 0.1 |
| Spring 14 | −1.0 | 0.3 | −0.9 | 0.3 | 0.6 | 0.6 |
| Summer 14 | −0.9 | 0.3 | −0.8 | 0.4 | 1.0 | 0.1 |
| Fall 14 | – | – | – | – | 1.0 | **0.0** |
| Winter 14 | 0.4 | 0.8 | 0.9 | 0.3 | 0.8 | 0.4 |
| Spring 15 | 0.4 | 0.8 | 0.1 | 1.0 | 1.0 | 0.2 |
| Summer 15 | – | – | – | – | −1.0 | 0.2 |
| Fall 15 | 1.0 | 0.1 | 0.8 | 0.4 | 0.9 | 0.3 |
| Winter 15 | 1.0 | **0.0** | 1.0 | **0.0** | 1.0 | **0.0** |

**Notes.**

in citizen-science camera trapping. In classroom settings, students learned about mammal species found in North Carolina and tested hypotheses they generated about which animals are most abundant in forested versus open areas at Prairie Ridge. The educational value of these programs is anecdotal but believed to be high, promoting awareness of mammals and camera trapping, as well as providing opportunities to participate in authentic and relevant scientific research to hundreds of people throughout North Carolina.

## DISCUSSION

These studies show that citizen scientist-run camera traps can be used to efficiently monitor mammal communities, address concrete management questions and suggest positive effects on volunteers. Where citizen scientists can be recruited to set cameras on their private lands, such as we did at Sandburg Home, citizen science offers access to areas where wildlife surveys would otherwise be impossible, allowing a more complete and representative sample. In addition, surveys of private lands may increase a sense of

**Table 6 Results of Fisher's exact test to determine significant differences in survey responses on a 5-point Likert scale between volunteers participating in camera trapping at Prairie Ridge and a control group, $\alpha = 0.05$.** Prairie Ridge volunteers expressed significantly greater interest in wildlife in general and urban wildlife, although significance was marginal.

| Question | $p$ |
|---|---|
| I like coyotes | 0.1 |
| I am interested in wildlife | **0.0** |
| I care about wildlife being able to live in urban areas | 0.1 |
| Seeing wildlife makes me happy | 0.4 |

**Notes.**

stewardship and empower landowners to take action, such as we found at Sandburg Home when private landowners produced newsletters to inform neighbors of our findings, put bear activity in context and offered advice to reduce nuisance bear encounters. In this way, volunteers previously acting in service to researchers became empowered as co-researchers, asking their own questions about ecology in their community and using the data for outreach. Citizen science monitoring at urban nature centers like Prairie Ridge also offers opportunities for public outreach to the community via citizen science ambassadors, garnering interest in urban ecology, urban wildlife and conservation. Finally, citizen science can leverage the necessary time and effort from volunteers required for long-term monitoring, something existing funds would not cover otherwise. Using these methods, we were able to gather sufficient data to meet management goals. Specifically, we were able to put bear activity into perspective at Sandburg Home showing occupancy and DR were similar to most adjacent sites. This finding has delayed any need for active management of the bear population at Sandburg Home, which had previously been considered due to a perceived high bear population. Compared to adjacent heavily forested sites, Prairie Ridge had overall higher mammal diversity and activity, especially of species associated with early successional habitats (i.e., eastern cottontail, white-tailed deer, coyote) (Figs. 2 and 7, Figs. S3, S5). This suggests that prairie management is having a benefit to biodiversity and relative abundance, supporting the continuation of this management action.

There is a growing need for long-term community datasets in ecology to help distinguish natural temporal changes from changes due to external factors (*Magurran et al., 2010*). Citizen science is a logical and attractive tool for small preserves that are unable to monitor their wildlife due to the lack of time and labor. Provided the sampling strategy and protocol are statistically sound and volunteers are not asked to operate outside of their comfort level, long-term monitoring by citizen scientists can provide valuable data. For example, at Prairie Ridge, year-round monitoring revealed seasonal trends in animal activity, such as when woodchucks entered and exited hibernation (Fig. 6). We noted very similar longitudinal patterns for detection of three species, raccoon, Virginia opossum and domestic cats that might correspond to suspected supplemental feeding schedules on an adjacent property, suggesting these species are leaving Prairie Ridge to access that food during those seasons, then returning (Fig. 6). Our monitoring also detected the appearance of infrequent visitors
to Prairie Ridge, such as bobcats, which are very rarely detected in Raleigh (Fig. 4). We found an increase in coyote activity near the beginning of the study which correlated with a sharp decrease in gray fox activity (Fig. 5), possibly indicating avoidance of coyotes by gray fox. The negative relationship between coyotes and gray foxes has been suggested by other studies, but is still poorly understood given the relatively recent eastward expansion of coyote range (*Chamberlain & Leopold, 2005*; *Neale & Sacks, 2001*). This information can later be used to help guide future management decisions and serves as important educational material for visitors.

Both case studies used DR (count/day) as a measure of relative mammal activity between sites, habitats, seasons and years. However, this method has been fairly criticized for not accounting for differences in detection probability which can vary both spatially and temporally (*Sollmann et al., 2013*), leading to potentially misleading results when used as an index of abundance or density (*Parsons et al., 2017*). However, when used as a measure of relative activity (i.e., intensity of use of a site/habitat), DR becomes less problematic and more similar to measures of occupancy in continuous habitat (i.e., use), particularly with careful study design and use of covariates to help control for movement rate differences between sites/habitats (*Parsons et al., 2017*). Although occupancy is advantageous and commonly used for monitoring because it does account for imperfect detection, managers without specialized training may find it daunting to use, and it is not particularly useful with very common species, as noted in this study with white-tailed deer. Using both methods simultaneously has been shown to give complementary information and a more well-rounded picture of animal activity from camera traps (i.e., *Kays et al., 2016*). We were able to account for differences in detection probability by modeling the detection portion of our occupancy model as a function of detection distance measured at each site, however, other unmeasured sources could have affected the differences in occupancy and DR we observed.

The potential educational benefits of citizen science to participants are wide-ranging from gains in knowledge of the natural world to hands-on experience with the scientific method (*Evans et al., 2005*; *Forrester et al., 2016*; *Jordan et al., 2011*). The educational goals at Prairie Ridge were met through mammal-themed programming and school visits. Based on comments from program participants and teachers and responses to our survey, citizen scientists learned about the role camera traps play in scientific research, discovered some of the mammal species in their areas, and improved their understanding of how urban habitat fragments such as Prairie Ridge benefit wildlife. Indeed, the benefits of hands-on work outdoors with the cameras became a valuable non-traditional education experience for some volunteers, exemplified by a student with autism who increased his communication and comfort level with others when doing field work with the cameras. The first few times this student worked with the cameras he was nearly silent and became visibly stressed especially when someone else in the group got to handle the camera. However, after several excursions camera trapping he increased his verbal communication with the group, telling stories about his experiences with his own camera traps and nuggets of personal information. He also became visibly more relaxed letting other people work with the cameras.

The Sandburg Home educational goals were more community-oriented, specifically to provide accurate, well-organized data to assist the community with planning efforts to address bear issues. We successfully engaged the local communities, which resulted in the dissemination of accurate information within those communities and beyond. We believe this type of engagement was possible because of the nature of the shared problem and the small, close knit community surrounding Sandburg Home. Based on comments in our signup sheet, most homeowners that participated in the study had experiences with bears in their neighborhood or property and were concerned for safety, or simply curious about the charismatic species. Anecdotally, homeowners seemed to know their neighbors well and readily communicate/socialize with them. If homeowners in the vicinity were having similar experiences with bears, this may have compelled participants to disseminate their new and relevant information resulting from study participation to their neighbors. The Sandburg Home staff plans to use the data gathered to develop a monitoring program and build a credible database to aid in the coordination of future wildlife conservation efforts with other concerned agencies and private landowners.

## CONCLUSIONS

Tools like camera traps that are easy to use, automated and produce verifiable data will continue to make more research feasible through citizen science (*McShea et al., 2015*). If special attention is paid to volunteer training and survey design, citizen science can be used not only in long-term monitoring, but to answer a variety of applied management questions at the same time promoting tolerance and curiosity about wildlife and the natural world (*Bonney et al., 2009*; *Dickinson, Zuckerberg & Bonter, 2010*). The ability to monitor over large areas for minimal cost is critical to the conservation and management of mammals, making citizen science an attractive solution. Coupling the ecological value of long-term monitoring with the educational value of citizen science offers the potential to reach communities anywhere biodiversity exists, both human and wildlife, in a variety of sensitive ecosystems exposed to anthropogenic change.

## ACKNOWLEDGEMENTS

We thank our 573 volunteers for their hard work collecting data for this study. We thank the staff of the Carl Sandburg Home National Historic Site and Prairie Ridge Ecostation. In particular we thank I. van Hoff for her assistance.

### Funding

This work was conducted with funding from the National Park Service [grant # P15AC00972], the US Forest Service [grant #13-JV-11330101-021], and the VRW Foundation. The funders had no role in study design, data collection and analysis, decision to publish, or preparation of the manuscript.

### Grant Disclosures

The following grant information was disclosed by the authors:
National Park Service: #P15AC00972.
US Forest Service: #13-JV-11330101-021.
VRW Foundation.

### Competing Interests

The authors declare there are no competing interests.

### Author Contributions

- Arielle Waldstein Parsons conceived and designed the experiments, performed the experiments, analyzed the data, contributed reagents/materials/analysis tools, prepared figures and/or tables, authored or reviewed drafts of the paper, approved the final draft.
- Christine Goforth performed the experiments, contributed reagents/materials/analysis tools, authored or reviewed drafts of the paper, approved the final draft.
- Robert Costello and Roland Kays conceived and designed the experiments, contributed reagents/materials/analysis tools, authored or reviewed drafts of the paper, approved the final draft.

### Human Ethics

The following information was supplied relating to ethical approvals (i.e., approving body and any reference numbers):

All survey protocols were approved by the North Carolina State University Institutional Review Board (protocol #11902).

### Field Study Permissions

The following information was supplied relating to field study approvals (i.e., approving body and any reference numbers):

A field permit was obtained from the US National Park Service to operate at Carl Sandburg Home National Historic Site. No permit was required to operate at Prairie Ridge Ecostation since staff were part of this research team.

### Supplemental Information

Supplemental information for this article can be found online at http://dx.doi.org/10.7717/peerj.4536#supplemental-information.

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
