# Peer review of "The value of citizen science for ecological monitoring of mammals"

_PeerJ, doi:10.7717/peerj.4536_

## Round 0.1 · original submission · Major Revisions

The paper presents two case studies as examples of citizen involvement in the process of science—here, animal monitoring and management. The reviewers note several instances in which a more compelling case can be made for the arguments presented regarding the value of citizen sciences. Some of this involves critical review of methods chosen and deployed (particularly concerned the DR method), a detailed discussion of results (including data entry and verification, data visualization for clarity), and more robust discussion that extends the understanding of citizen science as not simply crowd-sourcing data and should also include a discussion of the differences in the approaches used in each case study (and the role of the citizens). The reviewers note a variety of instances throughout the manuscript where the language is ambiguous and revision should carefully address each instance in which clarification is required. The raw data issues raised by Reviewer #2 should be carefully addressed, including the use license. Two reviewers raise objections to the inclusion of materials from other studies—Reviewer #2 provides pathways for inclusion and transparency to include these data. Authors should gain clearances for use of any images and data that are not generated by the study.

The decision to revise and resubmit was made due to the issues raised by the first reviewer. In balance with the positive comments from the second and third reviewers, the paper is certainly publishable but the major concern that must be addressed (and might require some extensive rewriting) is the consideration monitoring versus management—if the data are problematic due to the DR issues raised by Reviewer #1, are these case studies accurate models? Ultimately, the scope of the paper seems to be more about citizen science that using the data to understand animal behavior (and changes to it); if that is the case, Reviewer #3 provides clear pathways to refocusing the piece. But, one case study is argued to be a model for management so the concerns of Reviewer #1 must be addressed with regards to those statements. A revision should carefully consider which approach to take, and if the management issue remains, be careful about addressing the comments of Reviewer #1 in this regard. Otherwise, the detailed and helpful comments of the second and third reviewer, which constitute minor revisions, should be each addressed in turn.

Reviewer 1 ·

Basic reporting

The writing is clear and unambiguous and in general the paper conforms to a professional article structure. Raw data are provided and I could open the files.
The submission is not self-contained because it relies on data and analyses from other units of publication (Kays et al 2016, Parsons et al 2016).
The paper does not address hypotheses directly. Rather it uses 2 case studies to make the argument that citizens can collect data at an acceptable standard to answer management questions and monitor mammal activity. Mammal activity is not defined initially and, at first I thought the manuscript might be about activity rhythms, but it became clear that ‘activity’ referred to encounter rates or detection rates and that detection rates were ultimately used as an index of abundance. The first study seeks to quantify black bear activity inside Carl Sandburg Home NHS to help inform bear management in the area. They compare bear activity inside the NHS, near the NHS and to other sites in the region. The second case study aims to document how mammal species were using the Prairie Ridge nature center over time, including species interactions, and to determine the effect of active prairie restoration on the mammal community. There were education and outreach goals as well, including involving visitors in scientific research, improving knowledge of mammals, and improving understanding of urban wildlife.

Experimental design

Camera trap surveys: Both case studies use standard camera trapping protocols and attain reasonable intensity of sampling effort. The changes in sensitivity and camera models were documented. Both studies use detection rate (DR, defined as independent encounters) as the activity/abundance metric.
Volunteer recruitment and training: Adequate. The Prairie Ridge survey suffers from small sample sizes that make statistical comparisons and conclusions difficult.
Data collection and verification: There is little description of how data entry and verification was actually carried out.
Analyses: The analysis rests on use of DR to make comparisons of species spatially and temporally. The authors argue that use of DR is justified because sites were selected at random relative to animal movements and were not baited and cite Rowcliffe et al. 2013 as justification. Rowcliffe et al. 2013 is a clarification about assumptions underlying the random encounter model and does not make mention of detection rates. Strictly speaking the camera trap placements were not random because they used rules to avoid clustering the traps, and it is unclear whether they considered placement of cameras outside the target site on properties of volunteers only or on all properties in surrounding neighborhoods. Use of DR in place of abundance has been criticized when making spatial and temporal comparisons because detection probabilities can change over space and time while DR use assumes the DR remains constant. Several studies have shown, however, that DR can be a good surrogate for abundance, especially for species that are territorial and for species that move alone or in small groups.

Validity of the findings

Monitoring and management: The comparisons in Figure 2 confound spatial and temporal comparisons. There are many reasons for the observed differences but the authors only report that differences exist. Figure 2 stacks 15 species from Sandburg House in a bar chart that is unreadable. The results in Figure 3 for Sandburg Home bears report a significant difference but do not provide statistics or the nature of the test and multi-site comparisons are not described in methods. The Prairie Ridge result that DR was almost twice as high as forested sites nearby was not evaluated statistically and could arise due to a difference in detection zones – cameras set in forests tend to have smaller field of view that cameras set in open situations. Finally, the presentation of results for DR, trends over time, trends over space, and resident versus non-resident species in a single paragraph is confusing.
There is no discussion of the accuracy of classification by volunteers and of how much time was required by the scientists to clean up the data before archiving and use in analysis. There is discussion of management outcomes, only the results of monitoring.
Education and outreach: The Sandburg House results are not actually a part of the study as described in the methods and should be moved to the discussion. The results of the survey at Prairie Ridge are not derived from tests so we do not know if the changes described are significant rather than anecdotal.

Additional comments

While I believe many of the statements made in this manuscript regarding the effectiveness of citizen science and the widespread merits of involving citizens in the scientific process, this manuscript does not present a convincing argument. As I have pointed out, the methods have issues that require clarification and in the case of the interview survey, the sample sizes are too small to draw conclusions. The results do not necessarily apply to management because they do not get to the heart of monitoring – why changes are occurring. As reported, there are several possible explanations for why the results appear as they do, and the experimental design does not attempt to clarify which explanation is correct. The bear results are interesting but only help to inform public perception rather than management of bears. The title speaks of the value of citizen science to monitoring wildlife but does not describe that value other than that volunteers reduce the cost of doing science. This is a valid point, but one that has been made more strongly by some of the authors in Kays et al. 2016.

·

Basic reporting

I find this paper to be well-organised and easy to follow, and I am thankful for the opportunity to provide this review. The authors clearly described the need for more comprehensive ecological monitoring and how the challenges facing its implementation could be addressed through citizen science camera trapping. The figures and tables are also appropriate and helpful.

Some minor suggestions:

1. (Introduction lines 69-77) Citizen science is a broad term that is often misunderstood to only mean the crowdsourcing of data collection and classification. I suggest one or two sentences defining citizen science for the purposes of this paper while acknowledging its wider meanings for engagement and empowerment (such as, but not limited to, Haklay (2013): https://doi.org/10.1007/978-94-007-4587-2_7). While this study indeed focuses on data collection and classification, I think it is useful to say that citizen science is more than just that to prevent this misconception.

2. (Introduction line 98, Materials and Methods lines 124-125, Figure 1) Since this paper is being published in an international journal with a diverse audience, it should clearly report the study sites as being in the United States in the first instance. The map in Figure 1 should include an inset that puts the state of North Carolina in wider geographical context.

3. (Figure 2) I find it hard to read Figure 2 and differentiate the 16 species. It might help to separate it into two halves (e.g. Figure 2 a and b) that are identical except each half shows only eight species (i.e. eight species in each stack). This way, colours (and/or patterns) can be reused between the two and be chosen for greater contrast and colour-blind accessibility.

4. (attached dataset) I was able to access the raw data attached with submission of this article. Including the full dataset is a crucial part of ensuring research integrity and reproducibility and it is exemplary that the authors have done so. However, while most of the column names in the data spreadsheet are self-explanatory, some are not clear to me. I suggest including a “readme” file with the data describing the meaning of each column, how they are derived (if they are derived data), and their data types (e.g. integer, date (if so which format), alphanumeric string, etc.). Most importantly, the data cannot be legally used by others without an attached license. I suggest attaching the Creative Commons Attribution-ShareAlike license, or follow the data publishing guidelines here: https://opendatacommons.org/guide/

5. (Materials and Methods lines 211-212 and 223-224) The authors state that verified photos and data from previous projects are available in the Smithsonian and eMammalWeb repositories. Please provide information enabling access to those datasets including their Digital Object Identifiers (DOI), links for access, and license information.

6. (Discussion line 298) The word “manpower” is sexist, please find a synonym or rephrase.

7. The inclusion of animal silhouettes made the figures much easier to understand. The authors thoughtfully included links to the websites from which the silhouettes were obtained, and I followed the links to spot-check a few of the images. Some of them are shared under the Creative Commons Attribution-ShareAlike (CC BY-SA) 4.0 license, which legally requires attribution to the author (not just a link to the website) and a statement of the license in the attribution. For example, the chipmunk image (http://www.supercoloring.com/silhouettes/chipmunk) lists Natasha Sinegina as the author. This check should be done for all third-party material used and to ensure their licenses are legally compatible with the PeerJ Creative Commons Attribution (CC BY) 4.0 license (https://peerj.com/about/FAQ/#license).

Experimental design

The authors used two examples of citizen science wildlife monitoring with camera traps. They were well chosen for the temporal contrast between a short-term, “one-off”, project verses a longer term study, and also contrast in habitat (forest verses savannah-like). This way, results from these studies can better illustrate the breadth of applicability for the authors approach.

I believe the study – from site description, organisation of volunteers, education and outreach, to analysis – was well designed and described with a good amount of detail. For example, important information such as the number of photos per trigger, frame rate, and threshold for re-triggering of camera traps are presented. These important details are surprisingly and unfortunately often unreported in other papers.

A few responses and minor suggestions:

8. (Materials and Methods lines 198-200) To whom were the Twitter control surveys sent? Were they residents in the same area? Were the sample size and demographics comparable to that of the citizen scientists?

9. (Materials and Methods lines 207-209) The authors used eMammal to verify volunteer animal identifications. I am curious if the authors had to verify the classification of every single image or was it a spot check on a subset? What was the amount of photos needing verification and how substantial of an effort was this? If significant expert verification effort was required, it may be worth a bit of discussion as this may have implications for others seeking to conduct similar projects. In addition, the authors should link to the online repository for the full eMammal source code and associated license to ensure verifiability and reproducibility.

10. Following the previous point, is there any information on the natural history expertise (or any other pertinent demographics) of volunteers when they signed on to the project? Knowing whether skilled/knowledgable volunteers are needed may be useful for practitioners in the future.

11. (Materials and Methods lines 219-221) I note that the main statistical analysis reported in this paper is the Pearson’s correlation for seasonal correlations. As far as I can tell no other statistical methods were used for the other comparisons. I believe the current methods and results provide sufficient support for the authors’ conclusions, but I am curious if other statistical methods have been attempted or considered?

12. (Materials and Methods lines 219-221) The authors used the proprietary statistical software JMP. However, there were no supporting files attached with the submission and I was not able to review the analysis. It would be better if the relevant files (e.g. JMP project or analysis files) for the analysis are published with this paper under an open source license (https://opensource.org/licenses) such as as the GNU GPLv3 or later (https://opensource.org/licenses/GPL-3.0).

13. (Materials and Methods lines 223-224) I couldn't understand the last sentence in this section e.g. “with the except two at Prairie Ridge”. Can the meaning of this sentence be clarified?

14. (Results line 234) “Sandburg Home… had significantly lower bear activity compared to nearby Thurmond Chatham.” This is reasonably clear in Figure 3, but from which statistical method was this significant difference obtained?

15. While it can no longer be done now, I note that it would be ideal to conduct surveys and interviews on participants before the start of a project like this. This will establish a baseline on which possible changes in knowledge, attitude, and behaviour can be discovered.

Validity of the findings

The results and discussion are clearly linked to, and robustly derived from, the research questions in the introduction.

16. (Results lines 260-263) It is noteworthy that participants were empowered to initiate their own outreach, e.g. authoring “educational materials in the form of homeowner association newsletters...”. To me this stands out from other engagement outcomes where participants simply gained knowledge and/or skills. I believe it is worth emphasising in the discussion the potential of citizen science to empower participants in becoming more proactive citizens.

Additional comments

The needs for, and challenges facing, large scale ecological monitoring are great. While it is not the first time that citizen science camera trapping has been proposed to address this, the authors have conducted an important and comprehensive study on how such a project may be realised. This accessible paper is a solid and useful contribution to the discourse not just for ecologists, but for citizen scientists and conservation professionals (e.g. reserve managers or policy makers) as well. A few minor edits would further enhance its value.

Thank you for the chance for provide this review.

Reviewer 3 ·

Basic reporting

See general comments below.

Experimental design

See general comments below.

Validity of the findings

See general comments below.

Additional comments

(lines 80-82) Within the introduction, I’d like for the authors to note the type of citizen science methodology they are using (co-created, etc) and show familiarity with the different project types so as not to cast one view of what Citizen Science is.

(lines 83-84) Management of a large amount of data is not a new problem-- perhaps the authors could point to examples outside of ecology where big data is being successfully incorporated into decision-making processes (transit, urban design, etc. are good places to look).

(lines 87-88) I would like more information on the “expert review” function and how this supports verification of these large datasets on eMammal.

(lines 93-94) Again, I would like the authors to spend more time on a discussion of the types of citizen science they are undertaking. One case seems quite top-down while the other appear to answer questions from a community in close proximity to the site where they are working.

(Between lines 101-108) There are two very different types of projects being proposed-- one in which management practices will be implemented by the community and one in which they are implemented by the Center. It would be interesting if the authors could dig further into the differences of these approaches. Going more in depth on the citizen science aspect and the different methods and approaches is compelling, but not pulled out enough in this paper.

(lines 148-149) The article could use further explanation on why the cameras were left unchecked and (line 149) the reason for random site selection (perhaps moving lines 218-219 up or saying this earlier as well).

(line 197-200) Why not the same surveying approach to the Sandberg Homes group? If not surveying, what did the authors use to evaluate the approach with the Sandberg Homes group?

Beginning on line 253, because this is a relatively compelling use of citizen science to support community questions, I’d offer that the researchers should reconsider the use of terms such as “volunteers” and “participants”. These terms relegate people working on citizen science projects to being in service to the researchers, rather than empowered as co-researchers who are asking questions about their community and environment and also using the data to do things such as outreach to media and sharing their findings to neighbors via newsletters such as in the Sandberg Homes case. Although this is many times not valued as highly in citizen science, activities like this net highly relevant outcomes and should have higher placed value.

Line 265 notes there are 531 volunteers, but line 267 says only 9 completed the survey. Either this is written in a way that isn’t clear or more explanation needs to be given for the low survey numbers.

(lines 292-294) There is an argument to be had for citizen scientists driving down the cost of doing science, but by putting this in the forefront of the discussion section, potentially interesting information gained about educational and behavioral changes noted just above in the previous section is buried. Perhaps move the section that begins on line 318 up to meet the rest of the citizen science discussion. The chop in citizen science use analysis because of the discussion of what you discovered from an ecological perspective during the monitoring doesn’t quite flow.

The two sentences on lines 334-337 are incredibly compelling-- your work and approach is supporting people in creating their own planning efforts. This finding is buried and not built out, it would be great to see more focus on how and why this type of deep engagement in working towards community goals was successful in your example.

Lines 343-347, there is a lot of writing that is being done on this, especially from the citizen science and open technology fields. The assumptions and statements the authors make are accurate, but could be backed up with literature.

Charts and figures are clear-- would the authors consider including an image from the camera traps to demonstrate the raw data being used in this project or visual workflow of what citizen scientists are walking through when using eMammal?

---

## Round 0.2 · Minor Revisions

Thank for you for resubmitting your paper with a detailed response to the reviewers. This version appears to have addressed major concerns previously noted within the limits of the study. There are three issues that remain and must be fixed prior to acceptance.

1-language (see annotated PDF for instances of run-ons and ambiguous language)

2-statistics/tables
-format tables for readability and be consistent across tables
-include basic information for proper reporting of statistical analysis (e.g, test statistics) followed by p-value and include mean, SD, SE, confidence interval of mean (to evaluate significant and not significant results).
-limited each cell to contain one number (correlation results do not conform to this).
-all number columns should be right-justified
-A table header for p, can simply be p or Sig. That will reduce column width.
-please use full names for each test (Wilcoxon Signed-Ranks, not Wilcoxon method) on tables and in the paper. I did not flag this in the attached paper so please be sure that all tests are described using the proper name.
-include a table with the Shannon diversity analysis
-all tables with results should be primary tables, not supplementary, because they present primary analytical results and include significant findings.
-reference R and reference R package as noted in attached PDF

3-license and DOIs. See emammal links here and in attached file to address this:
-request DOI: https://emammal.si.edu/doi-request-emammal
-https://creativecommons.org/licenses/by-nc-sa/4.0/ which states that "You must give appropriate credit, provide a link to the license, and indicate if changes were made. You may do so in any reasonable manner, but not in any way that suggests the licensor endorses you or your use." You need to include a link to the license in any instance it is referenced or data or images from it are used. And DOI must be provided.

---

## Round 0.3 · accepted · Accept

Thank you for making the changes and going through the rounds of revision--the final piece is much improved and constitutes an interesting study showing the value of citizen science. I believe the paper is now ready for publication other than some minor lingering issues that I have flagged on the tables for the production staff (e.g., inconsistent decimal place within and across tables, inconsistent use of relative and absolute reporting of p-values). So look for those changes in the proofing materials.

Finally, you had commented how reporting of CI was not typical. Yes, it has not been in the past but most fields and journals are calling for evaluative statistics alongside test results. See here, if interested, for the American Statistical Association formal statement: https://www.amstat.org/asa/files/pdfs/P-ValueStatement.pdf. Now that we have online space for supplementary materials and are able to typeset tables for print publishing using digital information, we have the capacity to expand what we report to our readers.